# Obesity Paradox among Heart Failure with Reduced Ejection Fraction Patients: A Retrospective Cohort Study

**DOI:** 10.3390/medicina59010060

**Published:** 2022-12-28

**Authors:** Osama Abo Alrob, Sowndramalingam Sankaralingam, Sayer Alazzam, Buthaina Nusairat, Muhammad Qattoum, Mohammad B. Nusair

**Affiliations:** 1Clinical Pharmacy and Pharmacy Practice Department, Faculty of Pharmacy, Yarmouk University, Irbid 21163, Jordan; 2Clinical Pharmacy and Practice Department, College of Pharmacy, QU Health, Qatar University, Doha 2713, Qatar; 3Faculty of Pharmacy, Jordan University of Science and Technology, Irbid 22110, Jordan; 4Faculty of Pharmacy, Al-Zaytoonah University of Jordan, Amman 11733, Jordan; 5Department of Sociobehavioral and Administrative Pharmacy, College of Pharmacy, Nova Southeastern University, Fort Lauderdale, FL 33314, USA

**Keywords:** obesity paradox, heart failure, body mass index, diabetes, ejection fraction

## Abstract

*Background and Objectives:* There is consensus on the negative effects of obesity on the development of heart failure. However, several studies have suggested that obesity may have paradoxical survival benefits in heart failure patients. Therefore, the aim of this study is to investigate whether the obesity paradox exists in heart failure with reduced ejection fraction (HFrEF) patients in Jordan. Materials and *Methods*: In this retrospective cohort study, data were retrieved from electronic hospital records of heart failure patients admitted to King Abdullah University Hospital between January 2010 and January 2020. Patients were divided into five BMI (kg/m^2^) subgroups: (1) Less than 25.0, (2) Overweight 25.0–29.9, (3) Obese Class I 30.0–34.9, (4) Obese Class II 35.0–39.9, and (5) Obese Class III ≥40.0. Changes in patients’ clinical and echocardiographic parameters over one year were analyzed. *Results*: Data of a total of 297 patients were analyzed to determine the effect of obesity on heart failure. The mean age was 64.6 ± 12.4 years, and most patients (65.7%) were male. Among several co-morbidities, diabetes mellitus and hypertension were the most common and were present in 81.8% and 81.1% of patients, respectively. Over all patients, there was no significant change in EF after 1 year compared to baseline. However, only patients in the Obese Class I group had a statistically significant improvement in EF of 38.0 ± 9.81% vs. 34.8 ± 6.35% (*p* = 0.004) after 1 year. Importantly, among non-diabetic individuals, only Obese Class I patients had a significant (*p* < 0.001) increase in EF after 1 year compared to other BMI subgroups, a feature that was not observed among patients with diabetes. On the other hand, only Obese Class I patients with hypertension had a significant improvement (*p* < 0.05) in EF after 1 year compared to other BMI subgroups, a feature that was not observed among patients without hypertension. *Conclusions*: Our study demonstrates an inverted U-shaped relationship between BMI and EF such that patients with mild obesity (i.e., Obese Class I) had significant improvement in EF compared to those having a lower and higher BMI. We, therefore, suggest the existence of the obesity paradox among HFrEF patients in Jordan.

## 1. Introduction

Obesity has reached pandemic proportions, and it is estimated that more than 39% of the global population is either overweight or obese [1,2]. Obesity is a well-established major risk factor for the development of heart failure (HF) [3,4]. Obesity has independent adverse effects on cardiac structure and function [5]. The Framingham Heart Study has shown that for every unit increase in body mass index (BMI), the incidence of HF increases by 5% and 7% in men and women, respectively [3]. In addition, a recent causal genetic analysis suggests that obesity should be recognized as a causal factor for the development of HF [6]. This study shows that high levels of BMI causally increase the risk of incidence and mortality of HF. There has been increasing concern since, according to recent estimates, the prevalence of overweight and obesity in Jordan among adults is high, at 69.6% and 35.5%, respectively [7].

Although obesity is a risk factor for HF, over the last two decades, several studies and meta-analyses have shown better survival among obese patients with chronic HF in comparison to lean healthy or underweight patients; this is termed the “obesity paradox” [8,9,10,11,12,13]. This improvement in survival was associated with significantly higher left ventricular ejection fraction (LVEF) in overweight and obese subjects compared to underweight or healthy subjects, especially in patients with reduced ejection fraction (EF) [13,14,15]. Interestingly, the study that proposed a causal relationship between obesity and HF showed evidence of the obesity paradox using a case-only study design, in which obese HF patients had improved survival compared to HF patients with normal weight [6]. Furthermore, recent studies also support the obesity paradox for all-cause mortality; a J-shaped relationship was observed between BMI and risk of HF, with the highest risk in the morbidly obese group of patients [16]. A nutritional analysis of HF patients concluded that malnutrition resulted in a significant decrease in life expectancy, while obesity was associated with a significant increase in survival [17]. In contrast, another study showed that obese subjects with HF with reduced ejection fraction (HFrEF) had higher mortality risk than lean patients [18]. Thus, there is contradictory evidence on whether the obesity paradox is observed in both HF with preserved ejection fraction (HFpEF) and HFrEF.

Several hypotheses have been put forth to explain the paradox [19]. In obese HF patients, fat may serve as a metabolic reserve and serve as source of energy [20]. Increased lipid circulation binds to endotoxins in the obese, thus improving survival. The paradox may be due to early screening of obese individuals at a young age, which could lead to early diagnosis and treatment, conferring a better survival [21]. On the other hand, some investigators propose a “lean paradox”, in which HF patients with low body fat or low BMI may have poor cardiovascular outcomes [8,22]. While several of the above-referenced studies have discussed the effects of obesity on HF outcomes, intentional bariatric surgery-induced weight loss in obese patients was associated with a reduction in left ventricular hypertrophy and improvement in left ventricular diastolic function [16]. Moreover, significant weight loss induced by surgical treatment for obesity led to a reported 41% reduction in risk of HF [23]. In addition, a meta-analysis showed that intentional weight loss was associated with improvement in cardiac structure and function in obese patients [16]. Experimental studies from our group and others using animal models of HF have shown that lowering body weight in obese mice with HF improves cardiac structure and function [24,25]. While several studies support the concept of the obesity paradox in HF, some support the weight loss-induced improvement in cardiac function or survival as indicated above. The following points support our reasons for submitting in our study that an improvement in EF is a surrogate marker for survival. (1) It has been shown that higher LVEF is associated with improved survival among HFrEF patients [12,13,14]. (2) Furthermore, in patients with HFrEF, improved survival among the obese was associated with parallel and significant increases in LVEF [13]. (3) It has been shown that an increase in mortality rate is inversely proportional to LVEF [15]. More importantly, among patients with HFrEF, there is a linear relationship between decreasing EF and increasing mortality rates. Moreover, LVEF is an independent predictor of mortality in patients with LVEF ≤ 45%. Therefore, the aim of this study is to investigate whether the obesity paradox exists by analyzing LVEF in HFrEF patients in Jordan, as obesity is highly prevalent.

## 2. Methods

### 2.1. Study Design

This is a retrospective cohort study to determine the effect of BMI on EF in HFrEF patients over 12 months. Approval to conduct this study was obtained from the Jordan University of Science and Technology and the Institutional Review Board (IRB) at the King Abdullah University Hospital (KAUH), Irbid-Jordan, on 14 January 2021 (reference code 8/137/2021). Data were retrieved from electronic hospital records from patients admitted to hospital between January 2010 and January 2020 at KAUH. Unlike the majority of studies with similar objectives, in this study we assessed the association between BMI and EF and not survival or mortality because there was no mortality over the period of analysis among the selected subjects.

### 2.2. Inclusion Criteria

Inclusion criteria were as follows: HFrEF adult (18 years and older) patients with an EF < 45% on ECHO and increased left ventricular wall thickness and having complete follow-up medical records at KAUH over 12 months.

### 2.3. Exclusion Criteria

Patients less than 18 years old, patients with type I diabetes, and patients diagnosed with cancer, autoimmune disease, immune deficiency conditions during the 12-month follow-up period were excluded from the study.

### 2.4. Statistical Analyses

Data were analyzed using the Statistical Package for Social Sciences (SPSS^®^25). Categorical variables were expressed as numbers and percentages, and continuous variables as means ± SD. Data normality was assessed using the Shapiro–Wilk test and the *p*-value ≥ 0.05 indicated normally distributed data. The variables were assessed using a chi-square test or Fisher’s exact test (as appropriate) for categorical data, Students’ *t*-test for continuous data, and one-way ANOVA as appropriate followed by post hoc analysis. The difference between the groups was considered significant if the *p*-value was less than 0.05.

## 3. Results

### 3.1. Baseline Demographic, Clinical and Biochemical Parameters

Data from a total of 297 patients were analyzed to determine the effect of obesity on heart failure (Table 1). Overall, 65.7% (*n* = 195) of the patients were male, while the remainder, 34.3% (*n* = 102), were females. There was a significant difference in gender distribution within BMI categories (*p* < 0.001). A lower proportion of females was present in the lower three BMI categories compared to higher BMI groups. In addition, there was a significant difference in age between different BMI groups (*p* = 0.015). The Obese Class III patients were significantly younger by about 10 years compared to all other BMI groups.

Several co-morbidities were present among patients selected for this study. Diabetes mellitus and hypertension were the most common and were present in 81.8% and 81.1% of patients, respectively. Other co-morbidities, such as ischemic heart disease, chronic kidney disease or dyslipidemia, were present in about one-third of patients. There was no significant difference in co-morbidities among patients of different BMI categories. In addition, there was no significant difference in systolic blood pressure (SBP) among patients of different BMI groups. However, there was a significant difference in diastolic blood pressure (DBP) among patients in different BMI categories (*p* = 0.015). Patients in the lower three BMI categories had a lower DBP compared to those in the higher two BMI categories. Moreover, there were no significant differences in any of the lipid levels, such as total cholesterol, triglyceride, LDL, HDL or Hb_A1C_, among patients in different BMI categories. Furthermore, there were no differences in markers of renal function, such as creatinine and urea. Importantly, there were no differences in EF or LVWT among patients in various BMI categories at baseline.

### 3.2. Changes in Cardiac Structure and Function after One Year

For all patients taken together, there was no significant change in EF after 1 year compared to baseline (Table 2). However, patients in the Obese Class I group (BMI 30.0–34.9) had a statistically significant improvement in EF, at 38.0 ± 9.81% vs. 34.8 ± 6.35% (*p* = 0.004), after 1 year. EF among patients in BMI categories lower or higher than Obese Class I showed a marginal decline, thus suggesting an inverted U-shaped relationship between BMI and EF plotted on the x- and y-axes, respectively (Figure 1). However, changes in LVWT showed a different profile. Overall, there was a small but statistically insignificant increase in LVWT among all patients (*p* = 0.062). Importantly, only patients in the healthy range of BMI less than 25 had a significantly higher LVWT after 1 year (*p* = 0.045). On the other hand, among the overweight and obese groups, no specific pattern of change in LVWT was observed after 1 year.

### 3.3. Effect of Diabetes Mellitus or Hypertension on Changes in EF

Analysis was performed to determine whether the presence or absence of diabetes had any effect on EF (Table 3). Interestingly, among non-diabetic patients, there was a significant increase in EF after 1 year. This improvement was best observed in the Obese Class I category of patients, at 7.7 ± 7.89% (*p* = 0.033). A post hoc analysis revealed that the difference in EF observed in Obese Class I patients was significantly higher than all other categories of BMI. Patients with diabetes did not have a significant increase in EF after 1 year. On the other hand, patients with hypertension had significant improvement in EF after 1 year (*p* = 0.020). This increase in EF was highest for the Obese Class I group of patients and was significantly higher than the healthy, overweight and Obese Class II groups of patients, as observed in the post hoc analysis. On the other hand, patients without hypertension did not see an improvement in EF.

## 4. Discussion

The obesity paradox has been described to have a U-shaped relationship when increasing BMI is plotted on the x-axis and increasing mortality rate is plotted on the y-axis. This indicates that being overweight or mildly obese has a beneficial effect on survival. Our study investigated whether HFrEF patients in Jordan exhibit features of the obesity paradox, as characterized by a better survival or improvement in EF among obese patients compared to those with lower BMI. In our study, we identified three key findings. (1) One-year after baseline assessment, HF patients with Class I obesity had the best improvement in EF compared to those with lower or even higher BMI. (2) Class I obese HF patients without diabetes had the best improvement in EF compared to those with diabetes. (3) Hypertensive Class I obese HF patients had improvement in EF compared to those without hypertension. In summary, our study demonstrates an inverted U-shaped relationship between increasing BMI (on the x-axis) and increasing EF (on the y-axis) such that patients with mild obesity (Class I obesity) had significant improvement in EF compared to those having a lower and higher BMI (Figure 1). We therefore confirm the existence of the obesity paradox among HFrEF patients in Jordan.

Figure 1 is a representation of the average EF among patients in each BMI category. The connecting lines resemble an inverted U-shaped relationship.

Our data are consistent with a meta-analysis of individual patient data of 23,967 subjects. This study concluded that the obesity paradox was present in both HFrEF and HFpEF patients. This was characterized by a U-shaped relationship between BMI and mortality, with the lowest part of the curve showing Class I obese patients [13]. Furthermore, as part of the CHARM program, 7599 patients with HF were assessed for the obesity paradox. This study also observed that Class I obese HF patients had the highest survival rate [26]. The findings of our study, in which Class I obese patients had the maximal increase in EF after 1-year follow-up, are in close agreement with these studies. This is especially important because EF is a prognostic indicator of mortality in HF. Furthermore, a meta-analysis of nine observational studies concluded that being overweight or obese was associated with lower mortality in patients with congestive HF [12]. An inverse relationship between BMI and survival was observed. Several other studies have also reported a better survival among obese HF patients or increased mortality among low BMI patients, as reviewed in Nagarajan et al. [27].

Type 2 diabetes mellitus (T2DM) is a global epidemic; its incidence and prevalence have been on the rise over the last few decades [28]. HF is a common complication of diabetes, known as diabetic cardiomyopathy [29], and is more than twice as common among patients with diabetes compared to control subjects without diabetes [30]. Importantly, among hospitalized HFrEF patients, 42% had diabetes [31]. Thus, diabetes is significantly associated with HF. In agreement with above evidence, we observed that patients without diabetes had significant improvement in EF after 1 year. This effect was significant in the Class I obese group of patients compared to all other categories of BMI, further supporting the obesity paradox. One previous study that examined the impact of diabetes on HFrEF among 1930 patient pairs with and without diabetes mellitus concluded that those with diabetes had a poor prognosis and experienced increased length of ICU stay or hospitalization. They also had a higher risk of events such as cardiogenic shock and death during hospitalization [32]. In our study, HFrEF patients with diabetes had poor outcomes, whereas among those without diabetes, BMI influenced EF. Similar findings have been observed in other studies, in which BMI was a significant predictor of survival among non-diabetic patients with HFrEF. However, in diabetic patients with HF, BMI was not a significant predictor of survival [33,34].

In addition, the majority of patients with diabetes (76%) in our study did not have improvement in EF. Rather, they showed a small but insignificant decline in EF. Such an insignificant effect in our study could have been due to the fact that overweight or obesity co-existed in almost 71% of individuals, and therefore, the presence of diabetes may not have had an influence on EF over and above the effect of obesity itself. Although a previous study showed that women with DM had twice the increased risk of developing HF than men [35], we did not observe an effect of gender on EF. One study examined the effect of age on the obesity paradox. This effect was more prominent with increasing age, such that older individuals had better survival compared to younger patients for a given BMI [36]. However, our study did not find an association between age and changes in EF.

In contrast to the impact of diabetes on HFrEF, we observed that Class I obese patients with hypertension had improved EF compared to those without hypertension. This is interesting, since long-standing hypertension is a risk factor for HF [37]. Moreover, the absence of hypertension is associated with a lower life-time risk of developing HF. How hypertension impacts HFrEF is worth discussing. One study has shown that among older patients with HFrEF, having an SBP < 130 mm Hg was associated with a 7% 30-day all cause mortality compared to only 4% for those with SBP ≥ 130 mm Hg [38]. Patients with SBP < 130 mm Hg also had a higher risk of readmission for HF at 1 year compared to those with a higher BP. Furthermore, a recent study from China that examined the effect of diabetes mellitus on HFrEF showed that there was an increase in length of hospital stay in patients without hypertension compared to those with hypertension [32]. In addition, in symptomatic patients with systolic dysfunction, having lower BP was associated with greater mortality [39]. These data suggest a poor outcome for HFrEF patients with a lower BP. Rouleau et al. observed that the lower the pre-treatment SBP, the higher the risk of death among patients with HF [40]. Another study investigated the impact of hypertension on HFpEF patients. The authors observed that SBP < 120 mm Hg was associated with a higher risk of 30-day, 12-month and 6-year all-cause mortality compared to those with SBP < 130 mm Hg [41]. In the context of HF, hypertension could simply be an indicator of force of cardiac contraction and cardiac output, since BP is a measure of the force being exerted on the arterial wall when blood is ejected out of the left ventricle [37]. Therefore, it is intriguing to hypothesize that a higher BP could simply be an indicator of better cardiac performance and higher EF.

In our study, we observed that there was no significant change in LVWT after 1 year of follow-up among all patients taken together. However, there was a significant increase in LVWT only in patients with BMI < 25 but not in other higher BMI categories. This is in contrast to several studies that have shown an increase in LVWT in obese individuals compared to lean individuals [42,43] and that LVWT is positively correlated to BMI [44].

## 5. Conclusions

Our study confirmed the existence of the obesity paradox among HFrEF patients in Jordan. This has clinical implications in that physicians and healthcare teams treating these patients can better determine management strategies for weight loss and provide information to patients regarding their prognosis. Future studies will assess the obesity paradox in HFpEF patients.

## Figures and Tables

**Figure 1 medicina-59-00060-f001:**
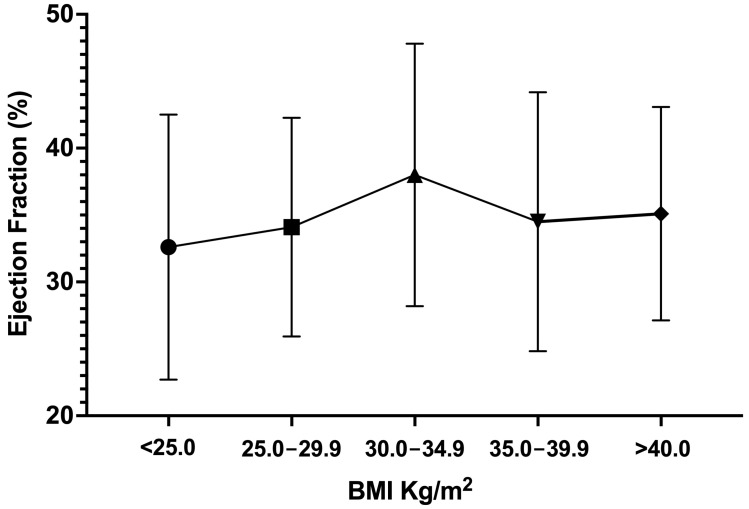
Representation of relationship between body mass index (BMI) categories and EF.

**Table 1 medicina-59-00060-t001:** Baseline demographics, co-morbidities and biochemical parameters.

Variable		All Patients	BMI Categories					*p* Value
			Less than 25 (*n* = 85, 28.6%)	25–29.9 (*n* = 93, 31.3%)	30–34.9 (*n* = 71, 23.9%)	35–39.9 (*n* = 28, 9.4%)	≥40 (*n* = 20, 6.7%)	
**Gender**	*Male*	195 (65.7%)	67 (34.4%)	64 (68.8%)	43 (22.1%)	10 (5.1%)	11 (5.6%)	**<0.001**
	*Female*	102 (34.3%)	18 (17.6%)	29 (31.2%)	28 (27.5%)	18 (17.6%)	9 (8.8%)	
**Age: mean (SD)**		64.6 (12.44)	64.8 (13.95)	65.1 (11.71)	65.8 (11.6)	66.0 (11.05)	55.4 (10.75)	**0.015**
**Co-morbidities**	*Diabetes Miletus*	243 (81.8%)	68 (28%)	79 (32.5%)	56 (23%)	25 (10.3%)	15 (6.2%)	0.581
	*Hypertension*	241 (81.1%)	72 (29.9%)	73 (30.3%)	59 (24.5%)	22 (9.1%)	15 (6.2%)	0.753
	*Ischemic Heart Disease*	87 (29.3%)	18 (20.7%)	33 (37.9%)	24 (27.6%)	8 (9.2%)	4 (4.6%)	0.204
	*Chronic Kidney Disease*	83 (27.9%)	29 (34.9%)	26 (31.3%)	18 (21.7%)	7 (8.4%)	3 (3.6%)	0.462
	*Dyslipidemia*	58 (19.5%)	13 (22.4%)	16 (27.6%)	21 (36.2%)	5 (8.6%)	3 (5.2%)	0.189
**Lab values**	*Systolic Blood Pressure*	127.9 (22.59)	125.0 (20.87)	126.7 (19.68)	128.8 (24.33)	134.6 (24.17)	132.4 (31.73)	0.288
	*Diastolic Blood Pressure*	76.0 (13.56)	74.1 (11.99)	75.6 (11.78)	74.9 (13.26)	80.8 (15.39)	82.6 (21.74)	**0.015**
	*Total Cholesterol*	3.5 (1.24)	3.6 (1.14)	3.3 (1.06)	3.8 (1.47)	3.2 (1.23)	2.7 (0.98)	0.244
	*Triglyceride*	1.6 (1.17)	1.5 (1.08)	1.5 (0.89)	1.9 (1.64)	1.4 (0.56)	1.6 (1.09)	0.591
	*Low Density Lipoprotein (LDL)*	2.2 (0.94)	2.3 (0.91)	2.1 (0.83)	2.3 (1.07)	2.0 (0.99)	1.5 (0.68)	0.318
	*High Density Lipoprotein (HDL)*	0.87 (0.32)	0.92 (0.32)	0.86 (0.31)	0.90 (0.32)	0.84 (0.38)	0.67 (0.23)	0.413
	HbA1c	7.8 (2.28)	7.6 (2.19)	7.9 (2.38)	8.0 (2.33)	7.2 (1.87)	8.9 (2.45)	0.172
	Creatinine	186.0 (142.69)	167.7 (142.68)	222.5 (166.43)	160.3 (93.93)	183.0 (133.04)	189.5 (157.85)	0.648
	Urea	15.9 (10.50)	13.9 (9.82)	16.7 (11.28)	16.0 (9.88)	18.4 (9.05)	16.6 (12.90)	0.263
	EF	34.4 (6.13)	33.2 (6.60)	34.6 (5.83)	34.8 (6.35)	35.6 (5.54)	35.8 (5.09)	0.595
	LVWT	1.11 (0.07)	1.10 (0.04)	1.10 (0.10)	1.10 (0.07)	1.10 (0.05)	1.12 (0.04)	0.995

**Table 2 medicina-59-00060-t002:** Effect of obesity on ejection fraction (EF) and left ventricular wall thickness (LVWT) after 1 year.

		Ejection Fraction (Time 1) Mean (SD)	Ejection Fraction (Time 2) Mean (SD)	*p* Value	LVWT (Time 1) Mean (SD)	LVWT (Time 2) Mean (SD)	*p* Value
**All patients**		34.4 (6.13)	34.7 (9.38)	0.557	1.11 (0.08)	1.12 (0.09)	0.062
**BMI groups**	**Less than 25**	33.2 (6.60)	32.6 (9.90)	0.507	1.10 (0.04)	1.12 (0.07)	**0.045**
	**25–29.9**	34.6 (5.82)	34.1 (8.17)	0.463	1.13 (0.10)	1.12 (0.20)	0.200
	**30–34.9**	34.8 (6.35)	38.0 (9.81)	**0.004**	1.11 (0.07)	1.12 (0.07)	0.096
	**35–39.9**	35.6 (5.54)	34.5 (9.67)	0.472	1.10 (0.05)	1.14 (0.06)	0.059
	**≥40**	35.8 (5.09)	35.1 (7.97)	0.595	1.12 (0.04)	1.12 (0.05)	1.000

**Table 3 medicina-59-00060-t003:** Effect of diabetes and hypertension on ejection fraction among different BMI categories.

		Ejection Fraction Changes Across Different BMI Categories Mean (SD)					*p* Value
		Less than 25 (*n* = 85, 28.6%)	25–29.9 (*n* = 93, 31.3%)	30–34.9 (*n* = 71, 23.9%)	35–39.9 (*n* = 28, 9.4%)	≥40 (*n* = 20, 6.7%)	
**Diabetes**	**Diabetic** **(*n* = 243)**	−0.4 (7.66)	−1.1 (7.37)	2.0 (8.92)	−1.2 (8.47)	−0.9 (6.08)	0.211
	**Non-Diabetic** **(*n*= 54)**	−1.0 (7.83)	2.4 (8.48)	7.7 (7.89)	−0.3 (3.21)	0 (2.74)	**0.033**
**Hypertension**	**Hypertensive (*n* = 241)**	−0.3 (7.49)	−0.2 (7.83)	3.7 (9.20)	−1.3 (8.60)	−0.7 (6.10)	**0.020**
	**Non-hypertensive (*n* = 56)**	−0.2 (8.66)	−1.8 (6.76)	0.7 (7.58)	−0.3 (6.12)	−0.6 (2.97)	0.861

## Data Availability

Available upon request from corresponding author.

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
