# Peer review of "Obesity Paradox among Heart Failure with Reduced Ejection Fraction Patients: A Retrospective Cohort Study"

_medicina, 2022, doi:10.3390/medicina59010060_

Round 1

Reviewer 1 Report

The study addresses a topic of interest. The population is appropriate since, as the authors state, the prevalence of overweight in Jordan is high. The article is well structured and the statistical methods are relevant. However, there are some formal aspects that should be improved to make the information more understandable to the reader.

1) Certain assertions (between lines 96 and 103) are not strictly speaking methods but reasons justifying that EF is a survival marker. I believe that this paragraph should be included in the introductory section, just before the objectives. 

2) In the statistical methods, comment that normality has been checked and that, depending on this test, Studen's t-test or Man Whitney U-test or ANOVA or Kruskall Wallis have been applied for the contrasts of means (I imagine that this is what they have done, but they do not indicate it).

Reviewer 2 Report

This study reports retrospective analysis of small cohort study of heart failure patients in Jordan. "Obesity paradox" was observed not only in human but in animal models of MI (unpublished material). It is obviously interesting subject and needs to be investigated further. Current paper presented statistical analysis on smaller number of patients  but it does not mean it has less value. Paper is easy to read and could be published in journal. 

Round 2

Reviewer 1 Report

The authors have incorporated the suggested changes, which have improved the quality of their work.